# Structured Spectral Graph Learning for Anomaly Classification in 3D Chest CT Scans

**Abstract.** With the increasing number of CT scan examinations, there is a need for automated methods such as organ segmentation, anomaly detection and report generation to assist radiologists in managing their increasing workload. Multi-label classification of 3D CT scans remains a critical yet challenging task due to the complex spatial relationships within volumetric data and the variety of observed anomalies. Existing approaches based on 3D convolutional networks have limited abilities to model long-range dependencies while Vision Transformers suffer from high computational costs and often require extensive pre-training on large-scale datasets from the same domain to achieve competitive performance. In this work, we propose an alternative by introducing a new graph-based approach that models CT scans as structured graphs, leveraging local slice triplets nodes processed through spectral domain convolution to enhance multi-label anomaly classification performance. Notably, our method exhibits strong cross-dataset generalization, and demonstrates competitive performance while achieving robustness to z-axis translation. We further conduct an ablation study to analyze the contribution of each component.

**Keywords:** 3D Medical Imaging · Chest Computed Tomography · Graph Neural Network · Spectral domain · Multi-label Anomaly Classification.

## 1 Introduction

Computed Tomography (CT) is a fundamental modality in modern medical imaging, providing radiologists with detailed cross-sectional views of the human body to detect and characterize abnormalities. However, the increasing volume of CT scans has led to an important demand for automated deep learning-based methods to assist radiologists with their growing workload [6]. Deep learning has already demonstrated success in various CT-related tasks [1], including anomaly detection [16], organ segmentation [21], image restoration [34], report generation [17], and synthetic volume reconstruction [16] for patient-specific modeling. Among these tasks, multi-label classification of anomalies in 3D CT volumes remains particularly challenging due to the computational complexity of processing volumetric data and the diverse range of pathological patterns. Early deep learning approaches leverage 3D Convolutional Neural Networks (CNNs),

effectively capturing local spatial features but suffering from limited capabilities to model long-ranges dependencies [24]. More recently, Vision Transformers (ViTs) [12], initially designed for natural language processing [30], have been adapted to both 2D [15] and 3D [18] medical imaging. By enabling long-range spatial interactions through self-attention, ViTs have shown promise in various medical imaging tasks [3] through its capabilities to capture global information. However, they remain computationally expensive, requiring large-scale pretraining to generalize effectively [18]. Our work introduces CT-Graph, a new GNN-based framework that models 3D chest CT scans as structured graphs, where each node represents a triplet of adjacent axial slices and edges are weighted by inter-slice spacing. This design enables efficient integration of local and global context while preserving spatial structure. Our approach offers the following key advantages:

- CT-Graph demonstrates strong cross-dataset generalization, maintaining consistent performance when trained on a public Turkish 3D chest CT dataset and evaluated on a separate dataset from the United States.
- Our edge weighting strategy based on z-axis distance spacing incorporates spatial awareness with no additional learnable parameters. Ablation studies confirm the effectiveness of GNN modules and graph connectivity patterns.
- By leveraging spectral domain convolution, CT-Graph improves anomaly classification performance and achieves robustness to z-axis translation.

## 2   Related Work

### 2.1   3D Visual Encoder

Feature aggregation in 3D medical imaging is crucial for balancing local and long-range dependencies while maintaining global spatial awareness. Early deep learning architectures primaliry relied on 3D CNNs [1], which effectively capture local spatial dependencies. These models have been widely applied to tasks such as anomaly detection [20] and segmentation [27]. However, their intrinsic locality limits their ability to model long-range dependencies, which can be crucial for capturing global anatomical structures [24]. The self-attention mechanism [31], initially introduced for natural language processing tasks was rapidly adapted to the visual domain with ViTs [12]. The extension of ViTs [18] and Swin Transformers [33] to 3D tasks has shown promise in applications such as dense image captioning [9] and video processing [23]. In the context of CT imaging, GenerateCT leverages CT-ViT, inspired by ViViT [2], to integrate spatial and causal attention but requires extensive pretraining, limiting its practical applicability [18]. To mitigate computational challenges in 3D volume processing, CT-Net [13] proposes to group triplets of adjacent slices to replicate the three-channel structure of RGB images, extracting features using a pretrained 2D ResNet [19]. While CT-Net subsequently passes these representations through a lightweight 3D CNN for dimensionality reduction, CT-Scroll [11] leverages an alternating global-local attention module to enable feature interactions, effectively reducing the number of parameters while improving classification performance.

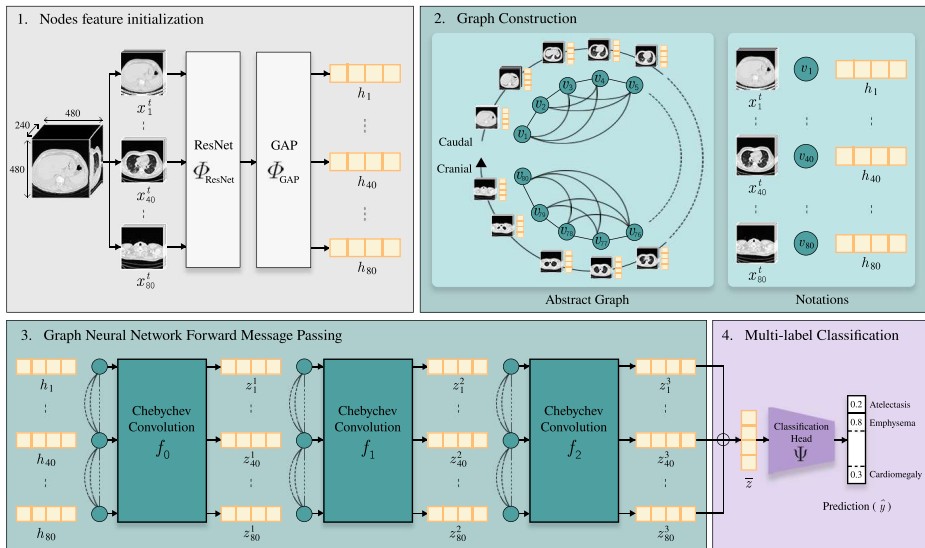

Fig. 1: CT-Graph introduces a structured graph-based architecture, where triplet axial slice features define nodes. Node interactions are modeled through spectral-domain convolutions, enabling contextual aggregation prior before classification.

## 2.2 Graph Neural Networks

In various application domains such as biology [28] or transportation [25], graphs are a common representation of data found in nature [32]. A graph, denoted as $\mathcal{G} = \{\mathcal{V}, \mathcal{E}\}$ consists of a set of edges $\mathcal{E}$ which model the connections between a set of nodes $\mathcal{V}$. In deep learning, GNNs have become the main approach for tasks involving graph-structured data [4], where each node is associated with a vector representation, which is iteratively updated through neighborhood aggregation during the forward message passing process. Representative models mainly include Convolutional GNNs, which aggregate neighboring node features through graph-based convolutions [10] or Attentional GNNs, which leverage attention mechanisms to weigh the importance of neighbors' contributions [7]. In medical imaging, GNNs have been used in tasks such as medical knowledge integration in radiology report generation [22] and Whole Slide Image analysis [14].

## 3 Method

As shown in Figure 1, CT-Graph models the 3D CT scan as a graph of *triplet axial CT slices* connected by their *physical z-axis distance*. Each node corresponds to a triplet of axial slices connected by neighborhood nodes with an edge weighted by their physical distance. Node features interact through a GNN module before being summed and given to a classification head.

**Triplet Slices Feature Extraction.** Following a strategy similar to CT-Net [13], we partition the input volume $x \in \mathbb{R}^{240 \times 480 \times 480}$ into non-overlapping triplets of slices, noted $\{x_i^t\}_{i=1}^{80}$ forming a tensor of dimension $80 \times 3 \times 480 \times 480$. Each triplet is processed by a ResNet [19] $\Phi_{\text{ResNet}}$ pretrained on ImageNet [29] to extract a corresponding feature map. The feature maps are then processed independently, with each one being passed through a Global Average Pooling (GAP) layer [11] $\Phi_{\text{GAP}}$ to obtain a compact vector representation for each triplet, noted $h_i \in \mathbb{R}^{512}$ ($i \in \{1, \ldots, 80\}$), such that:

$$h_i = (\Phi_{\text{GAP}} \circ \Phi_{\text{ResNet}})(x_i^t), \quad \forall\, i \in \{1, \ldots, 80\}. \tag{1}$$

**Graph Construction.** We define the volumetric representation as a graph $\mathcal{G} = (\mathcal{V}, \mathcal{E}, H, A)$, where:

- $\mathcal{V} = \{v_i\}_{i=1}^N$ is the set of nodes, where each node $v_i$ represents a triplet of consecutive slices. Hence, the number of nodes is $N = 80$.
- $\mathcal{E} \subseteq \mathcal{V} \times \mathcal{V}$ is the set of edges, where an edge $(v_i, v_j) \in \mathcal{E}$ is weighted based on a function of inter-triplet distance and z-axis spacing. An undirected edge $(v_i, v_j) \in \mathcal{E}$ is established if and only if the corresponding triplet slices are separated by at most $q \in \mathbb{N}^+$ other triplet slices in the sequence, such that:

$$\mathcal{E} = \{(v_i, v_j) \mid |i - j| \leq q\}. \tag{2}$$

- $H = \{h_1, \ldots, h_N\} \in \mathbb{R}^{N \times d}$ is the node feature matrix, where $\mathbf{h}_i \in \mathbb{R}^d$ denotes the feature embedding of node $v_i$ ($\forall\, i \in \{1, \ldots, N\}$). We set $d = 512$.
- $A \in \mathbb{R}^{N \times N}$ is the weighted adjacency matrix, where $A_{ij} = w_{i,j} \in \mathbb{R}^+$ encodes the connectivity and spatial relationship between triplets, $w_{i,j}$ being the edge weight such that:

$$A_{ij} = \begin{cases} w_{ij}, & \text{if } (v_i, v_j) \in \mathcal{E} \\ 0, & \text{otherwise.} \end{cases} \tag{3}$$

**Graph Neural Network module.** A key challenge in this formulation is the variability in anatomical positioning across patients due to differences in scan length and body proportions. Traditional spatial graph convolutions, such as GraphConv [26], aggregate information from fixed local neighborhoods, which can be suboptimal in this context as anatomical structures do not consistently align across scans. Instead, we leverage Chebyshev convolutions [10] to define graph convolutions in the spectral domain. Unlike spatial approaches, which struggle with non-uniform neighborhood structures [8], ChebConv utilizes polynomial approximations of the graph Laplacian [5] to capture hierarchical feature representations while preserving spatial localization. This allows the model to adapt to variations in caudal-cranial slice positioning and effectively learn long-range anatomical relationships, making it more robust to inter-patient variability. Our GNN module, denoted as $\Phi_{\text{GNN}}$, consists of 3 Chebyshev Convolutional Layers [10], each noted $f_n$ ($n \in \{0, 1, 2\}$), matching the depth of CT-Scroll [11] for fair comparison. For each layer, the scaled and normalized Laplacian $\hat{L}$ is defined as:

$$\hat{L} = \frac{2}{\lambda_{\max}}(D - A) - I, \tag{4}$$

where $\lambda_{\max}$ is the largest eigenvalue of the graph Laplacian $L = D - A$. The degree matrix $D$ is a diagonal matrix where $D_{i,i} = \sum_{j=1}^{N} w_{i,j}$. $w_{i,j}$ denotes the edge weight from source node $i$ to target node $j$, defined such that:

$$w_{i,j} = 1 + \frac{1}{1 + dist(i,j)} = 1 + \frac{1}{1 + 3 \times |i - j| \times s_z} , \tag{5}$$

where $s_z$ is the spacing along the z-axis in decimetre. The convolution operation is parameterized using Chebyshev polynomials $T_j(\hat{L}) \in \mathbb{R}^{N \times N}$, resulting in a recurrence relation for the transformation of the node feature matrix. Let $Z^0 = H$ be the initial node feature matrix, $\theta_k \in \mathbb{R}^{d \times d}$ be the learnable parameters, and $K$ be the Chebyshev filter size fixed to 3 for all experiments, to align with common practice [10]. The recurrence relation is given by:

$$Z^{n+1} = f_n(Z^n) = \sum_{k=0}^{K-1} T_k(\hat{L}) Z^n \theta_k, \quad \forall\, n \in \{0, 1, 2\} . \tag{6}$$

The GNN module $\Phi_{\mathrm{GNN}}$ produces the final output vector representation, which we denote as $Z = Z^3 \in \mathbb{R}^{N \times d}$ and which is defined as:

$$Z = \{z_1^3, \ldots, z_N^3\} = \Phi_{\mathrm{GNN}}(H) = (f_2 \circ f_1 \circ f_0)(h_1, \ldots, h_N) . \tag{7}$$

**Feature aggregation.** The obtained vector representations are aggregated through summation to derive a vector representation, denoted as $\bar{z} \in \mathbb{R}^d$, which is subsequently passed to a classification head $\Psi$ implemented as a lightweight multilayer perceptron. $\Psi$ predicts the logit vector $\hat{y} \in \mathbb{R}^{18}$. The model is trained on a multi-label classification task using Binary Cross-Entropy as the loss function.

## 4    Experimental results

### 4.1    Dataset preparation

We train and evaluate our methods on the public CT-RATE dataset [16], which consists of non-contrast chest CT scans with 18 annotated anomalies extracted from radiology reports. The training set includes 17,799 unique patients, while the validation and test sets both contain 1,314 unique patients. Additionally, we extend our evaluation on the publicly available Rad-ChestCT dataset [13], comprising non-contrast chest CT scans from 1,344 unique patients, focusing on the 16 anomalies shared with CT-RATE [16]. Consistent with prior work [17, 11], volumes for both datasets are center-cropped or padded to a resolution of 240×480×480, with a spacing of 0.75 mm on the x and y and 1.5 mm on the z axis. Hounsfield Unit values are clipped to the range $[-1000, 200]$, reflecting practical diagnostic limits [17].

### 4.2    Implementation Details

CT-Graph and baseline methods are trained with a batch size of 4 using the AdamW optimizer with $(\beta_1, \beta_2) = (0.9, 0.99)$ and a weight decay of 0.01. The learning schedule follows a cosine decay with a warm-up phase of 20,000 steps, a maximum learning rate of 0.0001, and training runs for 200,000 iterations.

Table 1: Quantitative evaluation on the CT-RATE and Rad-ChestCT test sets. Reported mean and standard deviation metrics were computed over 5 independant runs. **Best** results are in bold, second best are underlined.

| Dataset | Method | AUROC | Accuracy | F1 | Recall |
|---|---|---|---|---|---|
| CT-RATE | Random Pred. | $49.88\pm0.62$ | $49.89\pm0.31$ | $27.78\pm0.51$ | $50.42\pm1.05$ |
| | **ViViT** [2] | $79.19\pm0.28$ | $75.95\pm0.71$ | $49.91\pm0.28$ | $66.39\pm1.48$ |
| | **Swin3D** [23] | $79.94\pm0.15$ | $75.95\pm0.25$ | $50.64\pm0.25$ | $\underline{67.96}\pm0.58$ |
| | **CT-Net** [13] | $79.37\pm0.27$ | $77.37\pm0.40$ | $51.39\pm0.50$ | $66.42\pm1.99$ |
| | **CT-Scroll** [11] | $\underline{81.80}\pm0.22$ | $\mathbf{79.49}\pm0.45$ | $\underline{53.97}\pm0.21$ | $65.36\pm1.91$ |
| | **CT-Graph** | $\mathbf{82.44}\pm0.14$ | $\underline{78.66}\pm0.36$ | $\mathbf{54.59}\pm0.17$ | $\mathbf{68.77}\pm0.92$ |
| Rad-ChestCT | Random Pred. | $49.68\pm0.55$ | $50.40\pm0.32$ | $35.91\pm0.41$ | $51.51\pm0.75$ |
| | **ViViT** [2] | $67.83\pm0.38$ | $60.22\pm1.15$ | $48.59\pm0.97$ | $\underline{69.27}\pm1.64$ |
| | **Swin3D** [23] | $67.29\pm0.23$ | $60.67\pm0.60$ | $47.98\pm0.41$ | $66.76\pm0.63$ |
| | **CT-Net** [13] | $67.71\pm0.83$ | $60.05\pm1.93$ | $47.53\pm0.93$ | $68.45\pm1.18$ |
| | **CT-Scroll** [11] | $\underline{71.21}\pm0.37$ | $\mathbf{63.02}\pm0.93$ | $\underline{48.55}\pm0.54$ | $66.63\pm1.49$ |
| | **CT-Graph** | $\mathbf{72.18}\pm0.29$ | $\underline{62.60}\pm0.52$ | $\mathbf{49.52}\pm0.76$ | $\mathbf{69.30}\pm1.48$ |

## 4.3   Quantitative results

We evaluate model performance using standard classification metrics: AUROC, Accuracy and F1-Score (F1) which is the harmonic mean of precision and recall. For each method and each label, we select the threshold that maximizes F1-Score on the validation set and report all metrics on the test set. We compare our method against ViViT [2], a video-adapted Vision Transformer which also forms the architectural basis for CT-ViT, and Swin3D [33], an extension of Swin Transformer for volumetric data. We also include CT-Net [13] and CT-Scroll [11], two 2.5D approaches that employ CNN-based feature extractors. CT-Net relies on convolutional layers for feature aggregation and dimensionality reduction, whereas CT-Scroll leverages an alternating attention mechanism to capture cross-slice dependencies. ResNet-based models used ImageNet pre-trained weights; others were initialized via weight inflation [35] for comparability. Table 1 shows that CT-Graph consistently outperforms all baselines across AUROC, F1-Score and Recall. On the CT-RATE test set, our method achieves an F1-Score of 54.59, representing a $+\Delta1.15\%$ improvement over CT-Scroll [11] and $+\Delta5.93\%$ over CT-Net [13]. For the F1-Score, a paired t-test comparing the performance of CT-Graph against each baseline consistently yields a p-value $< 0.01$, demonstrating statistical significance. As shown in Fig. 2.a, CT-Graph yields the largest improvements on diffuse anomalies such as bronchiectasis, mosaic attenuation, and lung opacity. Reffering to Fig. 2.b, both attention and spectral convolution demonstrate robustness to z-axis translations, whereas standard convolution is sensitive to such shifts. To evaluate this property, we simulate patient body shifts by applying controlled translations along the z-axis with appropriate padding.

Fig. 2: (a) Per-anomaly F1-Score comparison for the 3 anomalies with highest improvement over baselines. (b) Model robustness to z-axis volume shift. F1 are reported for volumes translated along the z-axis with minimum-value padding.

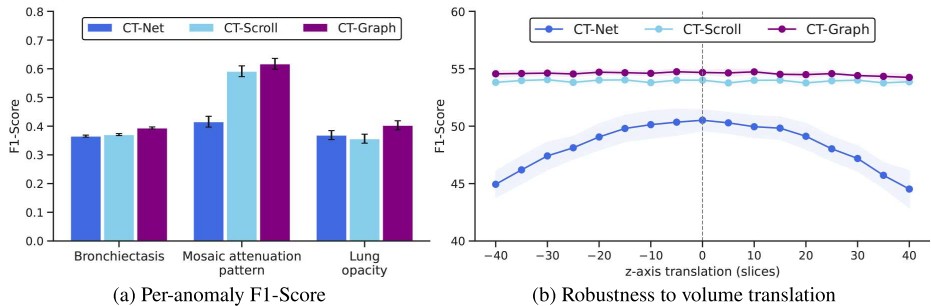

(a) Per-anomaly F1-Score              (b) Robustness to volume translation

| Connectivity | Module | AUROC | Accuracy | F1 |
|---|---|---|---|---|
| | GATv2Conv [7] | $81.56 \pm 0.03$ | $78.04 \pm 0.31$ | $53.72 \pm 0.34$ |
| *Fully connected* | GraphConv [26] | $81.99 \pm 0.40$ | $78.15 \pm 0.31$ | $53.73 \pm 0.36$ |
| | ChebConv [10] | $\underline{82.34} \pm 0.12$ | $\underline{79.01} \pm 0.55$ | $\underline{54.40} \pm 0.15$ |
| | GATv2Conv [7] | $82.22 \pm 0.05$ | $78.59 \pm 0.25$ | $54.06 \pm 0.19$ |
| *Neighbourhood* | GraphConv [26] | $82.33 \pm 0.18$ | $78.68 \pm 0.52$ | $54.16 \pm 0.24$ |
| | ChebConv [10] | $\mathbf{82.47} \pm 0.26$ | $\mathbf{79.12} \pm 0.53$ | $\mathbf{54.41} \pm 0.12$ |

Table 2: Comparison of graph connectivity schemes and GNN modules, evaluated on the CT-RATE test set. The neighborhood size is fixed to 16 for these runs.

## 4.4   Ablation study

**Comparison of representative GNNs.** Table 2 highlights the performance gains achieved by incorporating Chebyshev Convolutions [10] in our GNN module. Compared to a direct neighborhood aggregation approach [26], ChebConv improves AUROC by $+\Delta 0.42\%$ and F1-Score by $+\Delta 1.25\%$, suggesting that spectral-domain convolutions may enhance feature aggregation while demonstrating robustness to variations in cranial-caudal slice positioning (Fig. 2). Inference time takes approximately 70 milliseconds for all GNN variants.

**Graph construction.** Table 2 also shows that constructing a neighborhood graph leads to consistent improvements in AUROC and F1-score compared to a fully connected graph across all GNN variants. Table 3 also demonstrates that constraining the aggregation module's receptive field to a localized neighborhood improves anomaly classification performance, yielding a better balance between precision and recall. We adopt $q = 16$ as the optimal configuration.

Table 3: Impact of the neighbourhood size, using GraphConv as the GNN module. Neighborhood size, denoted as $q$, refers to the number of nodes each node is connected to via unweighted undirected edges.

| Neighbourhood size | AUROC | Accuracy | F1 Score | Recall | Precision |
|---|---|---|---|---|---|
| **4** | $\underline{82.22}\pm0.05$ | $\mathbf{78.97}\pm0.58$ | $\underline{53.76}\pm0.24$ | $66.02\pm0.92$ | $\mathbf{47.84}\pm0.22$ |
| **16** | $\mathbf{82.33}\pm0.18$ | $\underline{78.68}\pm0.52$ | $\mathbf{54.14}\pm0.24$ | $\underline{67.99}\pm0.75$ | $\underline{47.34}\pm0.30$ |
| **80** (Fully connected) | $81.99\pm0.40$ | $78.15\pm0.31$ | $53.73\pm0.36$ | $\mathbf{69.34}\pm0.91$ | $45.80\pm0.59$ |

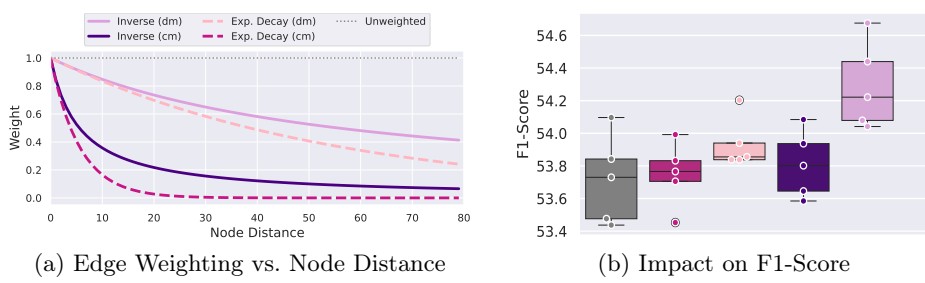

(a) Edge Weighting vs. Node Distance          (b) Impact on F1-Score

Fig. 3: Impact of the edge weighting functions, on the CT-RATE test set. We use a GraphConv module and a fully connected graph for all experiments.

**Impact of the weight function.** Among the evaluated edge weighting functions, the inverse function (see Eq. 5) with z-axis spacing measured in decimeters (dm) yields the best classification performance, as illustrated in Figure 3.

## 5    Discussion and Conclusion

In this work, we introduced CT-Graph, a new graph-based approach for multi-label anomaly classification from 3D Chest CT volumes. Each scan is represented as a structured graph, where nodes correspond to triplets of adjacent axial slices. To enable effective feature aggregation across this graph, we leverage a spectral approach based on Chebyshev convolution, which captures both short-range and long-range dependencies along the axial direction. Additionally, we show that incorporating spatially-aware graph structures, through both weighted edges and constrained neighborhood connectivity, enhances performance across multiple Graph Neural Network variants. CT-Graph demonstrates robustness to variations in patient body positioning along the z-axis and provides a flexible framework for modeling volumetric data. Future work may include anatomical segmentation-driven graph construction, transformer-based hybridization with mini-patch representations, and systematic exploration of architectural factors such as convolution depth and Chebyshev filter size.

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
