# OpenReview forum: "Structured Spectral Graph Learning for Anomaly Classification in 3D Chest CT Scans"
_MICCAI.org/2025/Workshop/MSB_EMERGE — MSB EMERGE 2025 Oral_

### Official Review · Reviewer_Tsyd · 2025-07-07

**Recommendation:** 3
**Confidence:** 4

**Clarity:**

The paper is generally clear but has some clarity issues that could be addressed with moderate revision

**Feedback:**

Except for the listed weaknesses:

-   The authors implement physical spacing (s_z) in their computation of the edge weights.
    However, during preprocessing all images are resampled to the same voxel spacing.
    The effect of this preprocessing step on the implementation of physical spacing
    could be investigated

**Justification:**

Missing related work and differences in model performance to the results reported in https://arxiv.org/pdf/2403.17834 are not clear.

**Reproducibility:**

Some amount of details available for reproducing the main results, and open access details are unclear

**Strengths:**

1.  The authors compare their method against four different SOTA model.
2.  Ablation studies investigating different parts of the proposed model are provided.
3.  The ability of graph-based models for classification of CT images is not studied in
    great detail yet.

**Summary:**

The authors develop a graph-based approach for classification of CT images. CT slices are encoded using a ResNet model pretrained on ImageNet, utilized as nodes. Edges of the graph are defined via the physical distance of different slices. The author train their method for classification of abnormalities on chest CT and compare their model against different state-of-the-art (SOTA) approaches, reporting superior performance. Ablation studies investigating different parts of the model are provided.

**Weaknesses:**

4.  Missing related work: In the related work section 2.2 the authors do not cite any
    other graph-based works that aim for classification of CT data.
    See, e.g.
    - Peng et al.: https://www.sciencedirect.com/science/article/pii/S1746809424001952
    - Kiechle et al.: https://link.springer.com/chapter/10.1007/978-3-031-83243-7_2
    Notably, Kiechle et al. also base their encoding on the relative position of
    the encoded slices.
5.  The authors compare their method against the CT-Net model trained on CT-RATE.
    In the original CT-RATE publication (https://arxiv.org/pdf/2403.17834) performance
    of CT-Net was also tested (see, Figure 4a in the CT-RATE paper).
    However, performances between the CT-Net model reported here and in the CT-RATE
    paper differ significantly, e.g. AUROC 79.37 vs. 62.9 and 67.71 vs 54.4.
    It is not clear where does differences stem from, as both methods should have
    been trained on the same datasets.

---

### Official Review · Reviewer_ecEZ · 2025-07-07

**Recommendation:** 4
**Confidence:** 4

**Clarity:**

The paper is clear and well-written, with minor areas for improvement in clarity

**Feedback:**

- For reproducibility reasons, it would have been nice to include a link to a public code repository
- Is there any way to not only include axial views but also views from other planes (saggittal, coronal) to support the claim of spatial-awareness?
- Instead of defining the edge-weights based on the slice distance, can they be learned from a complete graph using the weights of a GAT convolution operator?
- In case you used pytorch-geometric, please clarify how you incorporated edge weights for GATv2Conv as this is not natively supported out of the box
- You show that the way how you create the graph topology outperforms the fully connected scheme, but the improvement are very tiny, is this a significant finding?

**Justification:**

All in all it is a well structured and well written paper which shows a nice way to model slice dependence of 3D volumes as weighted graphs.

**Reproducibility:**

Some amount of details available for reproducing the main results, and open access details are unclear

**Strengths:**

The strengths of the presented paper are summarized as follows:
- elegant way to represent volumetric data as graphs, as this has been shown to be advantageous in an appropriate setting --> "Graph Neural Networks: A Suitable Alternative to MLPs in Latent 3D Medical Image Classification? - Kiechle et al."
- edge-weighting strategy based on z-axis distance spacing, which models local proximity of the slices as they appear in the input volume (novel contribution)
- With their method, the authors showed robustness to z-axis translations compared to baseline (CT-Net)
- evaluation of their method on multiple datasets (here two)
- clearly written and well structured
- graph topology evaluation, comparing their proposed method against a complete graph (fully connected), which is in close connection to how transformers are built
- Thorough ablation study to reveal the contribution of the models' building blocks

**Summary:**

The paper describes an approach which models tomography data (here CT) as structured graphs, to ultimately perform graph-level prediction on multi-class anomaly classification tasks. Therein the authors propose a graph edge-weighting strategy to preserve the spatial structure of the input data which is slice-wise (union of three consecutive slices) encoded using an 2D ImageNet pretrained ResNet encoder.

**Weaknesses:**

The weaknesses of the paper are summarized as follows:
- limited novelty, as the contribution "new graph-based approach that models CT scans as structured graphs" has already been published in literature --> "Graph Neural Networks: A Suitable Alternative to MLPs in Latent 3D Medical Image Classification? - Kiechle et al."
- Authors say that 3D CNNs can effectively capture local spatial dependencies but show limited ability to model long-range dependencies. Why was a simple 3D CNN not included as a baseline in the experiments in Table 1 to support that claim? (What is the Random Pred. method in Table 1? It is never addressed?)
- The experiments in Table 1 show an improvement of their proposed method against all baseline approaches, which is nice, but it is clearly missing how they compare in terms of trainable parameters. Is this a fair comparison?
- Authors compare the spectral graph convolution against spatial graph convolution and show that the performance is superior, but the ChebConv does have access to the information of three-hop neighbors (i.e., K=3), but the spatial approaches do only have the information of one-hop neighbors, which is unfair in comparison.
- In my opinion, the equation (5) in which you describe how the edge weights are computed does not belong to the subsection "Graph Neural Network module" but rather to the subsection "Graph Construction," which leads to confusion while reading the paper
- As the title of the paper shows, you are promoting spectral graph learning, why are you then doing some of the ablation studies only on the spatial GraphConv GNN? This does not make sense to me at all. --> Table3 and Fig3

---

### Official Review · Reviewer_ARMb · 2025-07-10

**Recommendation:** 3
**Confidence:** 5

**Clarity:**

The paper is clear and well-written, with minor areas for improvement in clarity

**Feedback:**

Already added with each comment.

**Justification:**

Paper needs a bit more work in terms of understanding the landscape of field and motivating the problem appropriately. Further, while the paper introduces a novel approach for 3D chest CT anomaly classification using spectral graph neural networks, it falls short in several critical areas such as insufficient technical depth and lack of more clinical insights.

**Reproducibility:**

Some amount of details available for reproducing the main results, and open access details are unclear

**Strengths:**

•	Paper is clear and easy to read.
•	Representing a CT volume as a structured graph of slice triplets is conceptually new.
•	The combination of spectral domain GNNs (Chebyshev convolutions) with z-spacing-aware edge weights is well motivated and technically novel.
•	Methodological details are well described, with clear equations and definitions. The ablation studies are comprehensive (GNN variants, neighborhood sizes, edge weights).
•	Evaluation across two datasets provides evidence of generalizability.

**Summary:**

This paper introduces CT-Graph, a novel Graph Neural Network (GNN) architecture for multi-label anomaly classification in 3D chest CT scans. The method models the scan as a structured graph where each node represents a triplet of adjacent axial slices. Edges are weighted using z-axis distance, and node features are updated using Chebyshev spectral convolutions. The method demonstrates strong performance across two datasets (CT-RATE and Rad-ChestCT), and exhibits robustness to z-axis translation.

**Weaknesses:**

•	While you acknowledge this, the triplet-based feature extraction heavily depends on a pretrained 2D ResNet. This limits the “volumetric” nature of the model. Consider discussing the implications of this 2.5D modeling choice in more depth.
•	Currently, the graph is constructed solely along the axial (z) axis. This misses potentially useful coronal/sagittal anatomical context or intra-slice relationships. However, you can discuss about the future improvements using anatomy-aware graph construction (you touch on this but could be stronger). Add a comparison table with model size and inference speed.
•	The qualitative analysis is limited. Include qualitative heatmaps or sample predictions to show what kind of anomalies the GNN captures better.
•	Even though the paper is about Graph Neural Networks, the literature survey about it is limited.
o	Kiechle, J., Lang, D.M., Fischer, S.M., Felsner, L., Peeken, J.C. and Schnabel, J.A., 2024, October. Graph Neural Networks: A Suitable Alternative to MLPs in Latent 3D Medical Image Classification?. In International Workshop on Graphs in Biomedical Image Analysis (pp. 12-22). Cham: Springer Nature Switzerland.
o	Kazi, A., Cosmo, L., Ahmadi, S.A., Navab, N. and Bronstein, M.M., 2022. Differentiable graph module (dgm) for graph convolutional networks. IEEE Transactions on Pattern Analysis and Machine Intelligence, 45(2), pp.1606-1617.
o	Ding, K., Zhou, M., Wang, Z., Liu, Q., Arnold, C.W., Zhang, S. and Metaxas, D.N., 2022. Graph convolutional networks for multi-modality medical imaging: Methods, architectures, and clinical applications. arXiv preprint arXiv:2202.08916.